# A Microfluidic Rotational Motor Driven by Circular Vibrations

**DOI:** 10.3390/mi10120809

**Published:** 2019-11-23

**Authors:** Suzana Uran, Božidar Bratina, Riko Šafarič

**Affiliations:** Laboratory for Cognitive Systems in Mechatronics, Faculty of Electrical Engineering and Computer Science, University of Maribor, Koroška c. 46, SI-2000 Maribor, Slovenia; suzana.uran@um.si (S.U.); bozidar.bratina@um.si (B.B.)

**Keywords:** microfluidics, micro-sized vortices in a water droplet, micro-sized rotational motor, piezoelectric actuators, circular vibrations

## Abstract

Constructing micro-sized machines always involves the problem of how to bring the energy (electric, magnetic, light, electro wetting, vibrational, etc.) source to the device to produce mechanical movements. The paper presents a rotational micro-sized motor (the diameter of the rotor is 350 µm) driven by low frequency (200–700 Hz) circular vibrations, made by two piezoelectric actuators, through the medium of a water droplet with diameter of 1 mm (volume 3.6 µL). The theoretical model presents how to produce the circular streaming (rotation) of the liquid around an infinitely long pillar with micro-sized diameter. The practical application has been focused to make a time-stable circular stream of the medium around the finite long vibrated pillar with diameter of 80 µm in the presence of disturbances produced by the vibrated plate where the pillar is placed. Only the time-stable circular stream in the water droplet around the pillar produces enough energy to rotate the micro-sized rotor. The rotational speed of the rotor is controlled in both directions from −20 rad/s to +26 rad/s. 3D printed mechanical amplifiers of vibrations, driven by piezoelectric actuators, amplify the amplitude of the piezoelectric actuator up to 20 µm in the frequency region of 200 to 700 Hz.

## 1. Introduction

The miniaturization of mechatronic devices, especially miniaturization of rotational micro-sized motors, has been at the forefront of the research efforts of scientists and engineers who have been developing microelectromechanical systems (MEMS) tirelessly for more than three decades.

The first micro-sized rotational motors (rotor sizes from 60 µm to a couple of hundreds µm) based on electrostatic energy supply to the rotor, were designed and tested in the laboratory environments in the years around 1990 [1,2]. They had controllable rotational speed (max. 50 rad/s) in both directions and a relatively high max torque in the order of pNm, but they were sensitive to the wear (life expectancy of a few hours [1]). The rotor was made from silicon.

An electromagnetic driven micro motor with rotor diameter 500 µm was reported in Ahn’s research [3] in 1993. It worked as a planar variable reluctance motor, and had fully integrated stator coils around the rotor with a diameter of 1.4 mm. The rotational speed was controlled in both directions. The same problems as for electrostatic micro motors were reported, with wear. The review paper reported the development of electromagnetic micro motors in the last two and a half decades [4], while review paper presented the methods for measurements of wear for micro electrostatic motors and some method for lubrication of their bearings [5].

The paper, published in the early 1990s, had reported a microfluidic motor with rotary gear trains on the silicon rotor (diameter 60 µm to 1600 µm) built into micro-sized channels to use fluidic linear stream of liquids for actuation as the driving force for gears [6]. The top speed was 390 rad/s with a high max torque of 8.7 pNm. The rotational speed of the rotor was controlled via the speed of the fluid flow in the tubes, and was not reported explicitly in the source. The problem of this type of micro motor was that it was closed in the micro-sized channel, and the rotational mechanical power could not be transferred outside the channel (pipe).

A rotational micro motor driven by electrostatic comb actuators was reported in Sniegowski’s research [7]. An exceptional rotational speed of up to 21,000 rad/s was presented, while driving multiple gears. The rotor diameter was 50 µm and it was 2.5 µm thick. There was no report about the max torque of the motor but, due to the low inertia of the rotor, the movement between the two equilibrium positions (steps) occurred on a sub-millisecond time scale. The device was operated at square-wave frequencies ranging from 0.5–3333 Hz, which corresponded to the rotational rotor speed 3–21,000 rad/s in both directions. There was also no report on the wear and durability of such a micro motor and its usage.

An interesting approach with piezo electric actuators was reported in Mashimo’s researches [8,9], where the rotor diameter was 700 µm and max torque was an astonishing 10 µNm. There were not any reports of wear. More information on other types of piezoelectric driven motors can be found in the review papers [10,11].

The source presented an unusual rotational motor with levitated pyrolytic graphite disk in a circular magnetic field [12], driven by a laser-emitted source of energy, with the rotational speed of 21 rad/s. The diameter of the disk was 3–10 mm. There was no report of max torque.

One of the first types of mm-sized rotor of a motor controlled by electro wetting was reported in Takei’s research [13]. The motor was composed of a 3.0 μL liquid droplet between two electric plates. The rotational speed of 19 rad/s of a rotor with a diameter of approximately 2 mm was achieved in both directions. Again, there was no report of max torque and wear.

The next micro motors, with micrometer to millimeter-sized diameters of rotors, used the energy of vibrations (surface acoustic waves, SAW) to drive a rotor floating in the liquid. Shilton et al and Yeo et al used two oppositely directed surface acoustic waves (frequency 20 MHz) driven fluid-coupled disk (diameter 5 mm) controlled in both directions at rotational velocities of 235 rad/s and at a max torque of 60 nNm [14,15,16]. They reported two versions: the first one needed liquid fluid between the disk and the substrate, and the other worked without the liquid. There was no information about the wear of these two types of motors. Kaynak et al used only one source of vibrations, at a frequency 4.3 KHz with amplitudes from 1–14 µm and specially designed rotors (diameter 600 µm) with 2–6 sharp edges [17]. Each sharp edge produced two whirls in the liquid, which rotated the rotor with max rotational velocity of 125 rad/s. The rotational velocity of the rotor around the pillar was controlled only in one direction. The max torque and possible wear for this application were not reported.

In the experiments [18,19,20], the whirls produced by the vibrations of the liquid rotated micro-sized living cells. The next experiments produced the whirls in a nanolitre-sized droplet by vibrations [21], or in the liquid film by an electric field [22]. The papers were the source of an idea used for the design of our type of micro motor [18,19,20,21]. The majority of the aforementioned vibrational micro motors [14,15,16,17,18,19,20], including electro wetting motor [13], used the design of the rotational disk or rotated living cell, which was not fixed in the space with an axis. This means that these approaches are not able to drive the gear-boxes. The only exception was the application with the rotor with the sharp edges [17], but this type of micro motor was not controllable in both rotational directions.

The overview of the micro motors discussed above is presented in Table 1.

The focus of our paper is to present a novel and unique microfluidic vibrational driven rotational micro motor, which produces the rotation of the rotor from the energy induced by the circular vibration in the water droplet. The paper presents the construction and control of a pillar-based vibrational microfluidic motor. The vibrational microfluidic rotational motor itself is a relatively simple device, consisting of a pillar in a droplet of water (liquid). Due to the circular vibrations, a circular water stream is created around the pillar in the water droplet, which drives the rotor of a microfluidic motor. Our type of micro-motor is driven by the circular vibrations produced by two perpendicular mounted piezo actuators with frequencies from 200 to 700 Hz and amplitudes of vibrations between 1–20 µm. A rotor with the diameter size 350–400 µm was mounted on the fixed pillar, allowing possible usage of gear-boxes. The rotational velocity of the rotor is controlled in both rotational directions from −20 rad/s to +26 rad/s. The max torque is estimated at 0.2 pNm with almost no wear, because the liquid in the bearings acted as a lubricant and decreased the wear. It is necessary to develop a high-quality vibration device to drive the water stream around the pillar. This article demonstrates the principle of operation, design, construction and testing of the pillar-based vibrational microfluidic rotation motor and its control subsystems necessary for quality and reliable operation. The advantages of the presented microfluidic motor are:(1)It uses the central axis, which fixes the rotor in both x and y directions on the top of the droplet and, therefore, allows mechanical connection to the gear box, which is not the case in sources [13,14,15,16,18,19,20].(2)It can work in a liquid environment, which allows the lubrication of the bearing between the pillar and rotor, and, therefore, decreases the wear, which is not the case in sources [1,2,3,4,5,7,8,9].(3)The rotor rotation can be controlled in both directions, which is not the case in sources [6,17].(4)It is useful for optical devices, because the components of the liquid motor, including the water droplet, can be made of transparent materials.

## 2. Materials and Methods

### 2.1. Materials

The polystyrene spheres (diameter 30 µm) were purchased from Kisker Biotech GMBH and Co. KG (Steinfurt, Germany). The vibrational pillars were made from aluminium or nickel wire with the diameter 50 µm, purchased from Alfa Aeser GmbH and Co. KG (Karlsruhe, Germany). Another type of pillar was made from optic fibre with the diameter 80 µm when the plastic envelope was removed from the fibre. The sensor for measurement of water droplet size was made from golden wire (Premion 99.995%, as a metal basis), with a diameter of 25 μm, purchased from Alfa Aeser GmbH and Co. KG (Karlsruhe, Germany). The micro-sized rotor was made from a quarter of a flattened Styrofoam ball, the glass-supporting plate (7 by 7 by 0.2 mm) was made from quartz glass, the mechanical amplifiers were made with a 3D printer from PLA filament. The piezoelectric actuators MPO Piezo Stacks MPO-050015 (resonant frequency (unloaded) f_res_ = 105 kHz, longitudinal piezoelectric large-signal deformation coefficient d_33_ = 10^−8^ m/V, el. capacitance at 1 V rms, 1 KHz C_p_ = 1,1 µF, maximal push force F_PUSH_ = 1500 N, maximal pull force F_PULL_ = 50 N, number of stages n = 10, stiffness k = 75 N/µm), from the company Nanofaktur (Villingen-Schwenningen, Germany), were used as a source of sine and cosine vibrations. The ultraviolet (UV) glue Bondic, purchased by the company ElektroG (Cologne, Germany), was used to fix the pillar to the supporting glass plate.

### 2.2. Laboratory Set-Up

The laboratory set-up (vibrating device) consisted of a vibration device mechanism (Figure 1a) and a vibration device signal generator. The vibration mechanism had a frame that is attached rigidly to the microscope (EUROMEX, Arnhem, Holland), and was capable of being moved left/right and up/down in the focus of the microscope lens. Two perpendicular mechanical amplifiers were connected rigidly to the frame. The glass plate was glued with the UV glue on the top of thin elastic connections between both mechanical amplifiers. The light beam of the microscope comes from the bottom of the microscope in the direction to the microscope lens directly above the pillar.

Both mechanical amplifiers were excited by their piezoelectric actuator MPO Piezo Stacks MPO-050015 by AC voltage signals with changeable frequencies (1–1000 Hz) and changeable amplitudes x and y (see Figure 1b) [23]. Both piezo actuators were excited with sine or cosine voltage signals of the same frequency, and the amplitude and the phase shift between both voltage signals could be changed independently. The piezoelectric actuators were located in the middle of the mechanical amplifiers, and were supplied with an AC voltage in the range −10 V to +115 V. This voltage range enables mechanical movements of the piezo actuator (x and y) with amplitude of approximately 10 μm. The mechanical amplifiers amplified the amplitude of vibrations x and y into a_1_ and a_2_, and changed the direction of vibrations by the angle of 90°. The tips of the mechanical amplifiers’ springs transferred the sine and cosine vibrations to the glass-supported plate, where circular vibrations were formed (see Figure 1b). Fastening screws were used to mount the piezoelectric actuators into the mechanical amplifiers. The piezoelectric actuators were voltage controlled with an Apex Microtechnology PA87 high voltage operational amplifier (Apex Microtechnology, Inc., Tucson, AZ, USA). The microscope was upgraded with the fast camera Pike (Allied Vision Technologies GMBH, Stadtroda, Germany). The microscope lenses (S. Plan M4x/0.10, ∞/0) and (S. Plan M10x/0.25, ∞/0) were used for observation of the microfluidic rotor rotation.

### 2.3. Methods

#### 2.3.1. Presentation of the Microfluidic Rotational Motor

The structure of the pillar-based, vibrating microfluidic motor is shown in Figure 2. The microfluidic motor fabrication started by applying a water-repellent liquid onto a clean and dried microscope glass, and then polishing the lubricated surface. In this way, the glass surface became hydrophobic and repelled a drop of water. Consequently, the drop of water, located around the axis of the motor–pillar, has a distinct and stable circular shape at the top. The pillar (motor axis) was made of a thin glass cylinder with a diameter of 80 μm and a length of 2–3 mm and was glued perpendicularly to the glass base with UV glue. The glass cylinder was made from a glass optical fibre after removal of the plastic envelope. The rotor of a microfluidic motor is a disc lighter than water with a hole in the centre and a diameter of approximately 350 μm. The disc was made from a Styrofoam ball (approximately 350–400 μm in diameter). The Styrofoam ball was cut three times and a part of approximately one quarter of the ball was deformed thermally (flattened) with two metal surfaces heated to approximately 70 °C for a few seconds. Then, the hole, with a diameter of approximately 100 μm, was made with sharp needle tool. The disc (rotor) was put on the pillar on the top of the water droplet. The disc floats freely on top of the droplet and only flattens the droplet top slightly.

The rotation of the water stream in the droplet around the pillar, and, consequently, rotation of the disc (rotor) of the microfluidic motor around the pillar, is caused by the circular vibration of the glass-support plate. The formation of the microfluidic circular stream of liquid around the pillar with micro-sized diameter is described in detail in the next subsection.

The constructed microfluidic motor is shown in Figure 3.

#### 2.3.2. Microfluidic Streaming of Water around the Vibrating Pillar

The mathematical model of vibration-induced whirling flow of incompressible fluid around the infinitive long pillar with micro-sized diameter and its development were presented in [18,19,24]. We followed the model development from [18], and obtained slightly different equations with almost the same results, while the mathematical model from [19] is wrong, probably a typewriting error, despite that the authors used the same development approach. So, our developed model for the steady term of $\psi_{st}$ is:(1)$$\begin{array}{ll} \psi_{st}= & r^{4}\left(\frac{1}{48}\int_{a}^{r}\frac{1}{x}\rho \left(x\right)dx+c_{1}\right)+\;r^{2}\left(-\frac{1}{16}\int_{a}^{r}x\rho \left(x\right)dx+c_{2}\right)\\ & \;\;\;\;\;+\left(\frac{1}{16}\int_{a}^{r}x^{3}\rho \left(x\right)dx+c_{3}\right)\;+\;\frac{1}{r^{2}}\left(-\frac{1}{48}\int_{a}^{r}x^{5}\rho \left(x\right)dx+c_{4}\right) \end{array}$$
where
(2)$$\rho \left(x\right)=\frac{2\pi^{3}f^{3}A^{2}}{\eta^{2}}\left[2Y+2Y^{*}-2\frac{a^{2}}{r^{2}}CD-2\frac{a^{2}}{r^{2}}C^{*}Z-4YY^{*}+4ZZ^{*}\right]$$
and
(3)$$c_{1}=-\frac{1}{48}\int_{a}^{\infty }\frac{1}{x}\rho \left(x\right)dx$$
(4)$$c_{2}=\frac{1}{16}\int_{a}^{\infty }x\rho \left(x\right)dx$$
(5)$$c_{3}=\frac{a^{4}}{16}\int_{a}^{\infty }\frac{1}{x}\rho \left(x\right)dx-\frac{a^{2}}{8}\int_{a}^{\infty }x\rho \left(x\right)dx$$
(6)$$c_{4}=-\frac{a^{6}}{24}\int_{a}^{\infty }\frac{1}{x}\rho \left(x\right)dx+\frac{a^{4}}{16}\int_{a}^{\infty }x\rho \left(x\right)dx$$
where *f* is the frequency and *A* is the amplitude of the applied vibration, *r* is the distance from the pillar, η is the kinematic viscosity of the fluid, and *a* is the radius of the pillar. The functions *Y*, *Z*, *C* and *D* were calculated from the Hankel functions of the first kind $H_{n}^{\left(1\right)}$ and from Hankel functions of the second kind $H_{n}^{\left(2\right)}$ with the next equations: (7)$$Y=\frac{H_{0}^{\left(1\right)}\left(\varepsilon .x\right)}{H_{0}^{\left(1\right)}\left(\varepsilon .a\right)},Z=\frac{H_{2}^{\left(1\right)}\left(\varepsilon .x\right)}{H_{0}^{\left(1\right)}\left(\varepsilon .a\right)},C=\frac{H_{2}^{\left(1\right)}\left(\varepsilon .a\right)}{H_{0}^{\left(1\right)}\left(\varepsilon .a\right)}\;,D=\;\frac{H_{2}^{\left(2\right)}\left(\varepsilon z.x\right)}{H_{0}^{\left(1\right)}\left(\varepsilon .a\right)}$$
(8)$$\varepsilon ={\left(i\frac{2\pi f}{\eta }\right)}^{\frac{1}{2}}\;,\varepsilon z={\left(-i\frac{2\pi f}{\eta }\right)}^{\frac{1}{2}}$$
Finally, we got a tangential velocity $v_{t}$ and a rotational velocity $v_{r}$ of the liquid stream around the pillar with:(9)$$v_{t}=\frac{\delta \psi }{\delta r}\;,\;v_{r}=\frac{1}{r}\frac{\delta \psi }{\delta r}$$
The Matlab code of the above mathematical model is attached as Appendix A.

The rotational speed $v_{r}$ (rad/s) of the rotated water stream around the pillar against the distance from the pillar r is shown in Figure 4. The liquid was water at 20 °C (kinematic viscosity η= 1.0 × 10^−6^ m^2^/s), the radius of the pillar was 40 µm, the amplitude and frequency of the circular vibration were A = 18 µm and f = 530 Hz, respectively. The rotational speed is the highest at the distance r = 36 µm away from the pillar. So, with regard to the slope of the curve in Figure 4, the practical rotor radius of the motor should not exceed the value 150 to 200 µm, otherwise the average rotational speed of the water stream around the pillar and, consequently, the rotational speed of the rotor (average rotational speed $v_{r}$*)* would be reduced dramatically. The average rotational speed was calculated under the assumptions that the roughness of the rotor surface in contact with the water was the same in the whole surface, and the whole rotor surface was in contact with the water droplet surface. Equations (1–9) showed that the rotational speed $v_{r}$ can be increased by increased values of frequency *f*, amplitude *A* and decreased value of the fluid’s kinematic viscosity η [18,19,24]. Demonstration of the rotating of a water stream (polystyrene sphere with diameter 30 µm) around the pillar in the shape of a glass sphere (diameter 100 µm) glued to the support glass plate is presented in Appendix A.

#### 2.3.3. Microfluidic Rotating of Water above the Vibrating Glass Plate

The rotation of the water stream around the pillar with micro-sized diameter was not the only rotation of water in the droplet. The glass-support plate also induced its own rotation vortices. The more or less random distribution of rotated water stream vortices was presented only when the circular vibrations with micro-sized amplitudes had excited a thin layer of water above the glass-support plate. Several experiments were performed to prove the existence of the vortices. The frequency and amplitude of circular vibrations were changed between 1 Hz < f < 1000 Hz and 1 µm < A < 10 µm, respectively. The depth *d* of the water layer in a small pool with dimensions 10 by 20 mm was changed between 1 mm and 10 mm. The smaller the depth layer was, the more randomly distributed vortices were observed on the surface of the water. The vortices disappeared from the water surface when the depth of the water in the pool increased above the level of 8 mm. The number, diameter, the place of vortices, the speed and direction of rotation, had changed during the changed frequency, amplitude and depth of the water layer. Thus, we treated the presence of these vortices as a disturbance when they appeared in our water droplet. Appendix A shows the rotation of water above the glass plate in the pool at frequency f = 300 Hz, A = 10 µm and depth d = 1 mm. Appendix A shows the rotation of water above the glass plate in the pool at frequency f = 300 Hz, A = 10 µm and depth d = 3 mm.

#### 2.3.4. Mechanical Amplifier

The principles of mechanical amplifiers used for our device were presented in Lin’s research [23]. Several types of mechanical amplifiers exist, but only this one offered us static amplification of vibrations (at frequency 0 Hz) with ratio 3 and resonance amplification (around 500 Hz) with ratio up to 26. The scheme of the mechanical amplifier is shown in Figure 5. The piezo actuator (MPO Piezo Stacks MPO-050015) was installed in the middle of the mechanical amplifier, and was clamped with a screw between holders. A clamping screw was used to mount the piezo actuator precisely between the holders, so there was no backlash during vibrations. The spring is only half a millimetre wide with a 5 mm height connection, and is used to decrease as much as possible the mechanical coupling between both mechanical amplifiers across the glass-supporting plate. The mechanical amplifier is attached to the frame with two screws, while the glass-support plate was attached (glued) to the tip of the spring. The mechanical amplifiers used in our application were made by 3D printing from PLA plastic material. We had also tried other materials (ceramic, aluminium), but only the PLA plastic with a honeycomb internal structure allowed the mechanical amplification ratio up to 26 at the resonant frequencies. The dimensions of the mechanical amplifiers were the same as in Lin’s research [23]. The only difference was our innovation to add a clamping screw, which allowed easy replacement of the piezoelectric actuator and decreasing the backlash during vibrations. The amplification ratio of 26 at the resonant frequency was needed because a high voltage operational amplifier PA87 decreases the output voltage amplification higher frequencies. The piezo actuator with only electrical amplification, without the mechanical amplifier, had an output amplitude at approximately 500 Hz only 2–3 µm. This small amplitude was not enough to induce the water rotating around the pillar in the water droplet.

#### 2.3.5. Measurements of Vibration Amplitude and Phase Shift between the Directions of Vibration Amplitude

The vibrations’ shapes, measured with a low weight accelerator probe, were distorted heavily by the weight of the accelerator probe. The reason is the weight of the probe, which was several times heavier than the weight of the glass plate, including the water droplet and the pillar. So, another method was used for measurement of amplitude of vibrations and the phase shift between the directions of the vibrations’ amplitudes.

The useful method for measurement of amplitude and phase shift of circular vibrations was based on the observation of the micro-sized polystyrene sphere with diameter 30 µm glued on the upper surface of the glass-support plate (see Figure 6a). First, the diameter of the sphere d was measured when both amplifiers were switched off. Second, only one of both amplifiers (i.e., first amplifier) was switched on, and the distance g was measured (see Figure 6b). The distance g presented the “stretched shadow shape” of the sphere, caused by capturing the picture with the camera in the direction of the vibration due to actuation of the first amplifier. Next, the other, (second) amplifier was switched on, while the first one was switched off—the distance f was measured (see Figure 6c). If both amplifiers were switched on, then the shape of the sphere was “stretched” in all directions. The vibration amplitudes due to amplifier 1 a_1_, and due to amplifier 2 a_2_, were calculated by:(10)$$a_{1}=\frac{g-d}{2}\;,\;a_{2}=\frac{f-d}{2}$$

In the end, phase angles (phase shifts) *α* were measured between the direction of distances *g* and *f* across the centres of both stretched shapes. The phase shift between vibrations was either angle 0° < α < 180° or 180° < α < 360. The phase angle 0° < α < 180° caused the rotation of water around the pillar in the clockwise (CW) direction, while the phase angle 180° < α < 360° caused the rotation of water in the droplet around the pillar in the counter clockwise (CCW) direction. The measurement error during the measurements of values *d*, *g* and *f* were about 1 µm (the resolution of the optical microscope), while the shift angle *α* had a maximal error of about 5°.

#### 2.3.6. The Volume of the Water Droplet

The volume of the water droplet during the operation slowly decreased or increased because of evaporation or condensation of water from the surrounding air. The volume, or better, the height of the droplet, was important to achieve constant operational conditions of the presented microfluidic motor. Water from the water droplet decreased its volume by one-third due to evaporation in half an hour of operation at temperature 20 °C and relative humidity of the surrounding air 30%. Such loss of water droplet volume stopped the operation of the microfluidic motor completely. Also, if the condensation increased the volume, for example by 4 times, then the operational parameters (frequency, amplitude needed for controlling the speed and direction of disk rotation) of the microfluidic motor deteriorated to the level of failure. So, precise control of the volume was needed for the proper operational exploitation of the motor. Two rings of golden wire (diameter is 25 µm) were glued to the glass with the pillar in the centre. Both rings were connected with a comparator. The comparator measures the resistance between both rings. If the water from the water droplet made contact between both golden wires, then the resistance was small in comparison with when the resistance was huge in the case where the water did not make contact. The first ring had the diameter 0.75 mm, while the outer one had the diameter 1.0 mm. If the bottom diameter of the water droplet was smaller than the inner ring with diameter 0.75 mm, then the cooled air came from the nozzle and cooled down the supporting glass plate from the bottom. The air from the nozzle was cooled down with a Peltier element to a temperature lower than the dew point temperature (approximately 8 °C) for the surrounding air at 20 °C and RH = 30%. When the supporting glass reached the temperature below the dew point temperature, then the water droplet volume started to increase due to condensation, and vice versa when the water droplet volume increased so much that its bottom diameter reached the outer ring with diameter 1.0 mm, then the temperature of the Peltier was increased to above the dew point temperature. The supporting glass plate started to heat up, and, consequently, the water droplet volume started to decrease due to evaporation. The time of condensing water from the surrounding air in which the bottom of the water droplet increased from inner ring to outer ring was 2–3 min. The comparator switched the Peltier element’s reference temperatures from below to above the dew point temperature plus the hysteresis of 3 °C.

## 3. Results

The previous section described the method of microfluidic motor rotation and disturbances (vortices due to the glass-supporting plate, distorted vibrations due to electrical and mechanical amplification, changed water droplet volume due to evaporation/condensation) which deteriorated, or even prevented, the operation of the microfluidic motor. The conditions for proper operation of the microfluidic motor and the results are shown in this Section.

### 3.1. Microfluidic Streaming of Water in the Droplet

Different types (size, shape, place of centre) of vortices were found during the tests. Some of the vortices’ centres were placed in the vertical axis of the pillar; and some were placed far away from the pillar. Sometimes, there were two, or even more vortices with different diameters in the water droplet, and the directions of water stream in the vortices were, at the same time, different. Some of the most frequent types of actuating vortices and their causes (the direction and amplitude of vibrations due to each mechanical amplifier) are presented in Figure 7. A single circular vortex around the pillar with micro-sized diameter, caused by two vibrations of the same frequency and the same amplitudes *a*_1_ and *a*_2_ and phase shift α = 90°, is shown in Figure 7a. This kind of vortex was predicted by the theoretical equations presented in Section 2.3.2, but it was not often seen in the laboratory experiments. If the amplitude *a*_1_ > *a*_2_ and the phase shift α = 90° then two circular vortices appeared, the larger one centred to the pillar, while the smaller one was placed nearby the water droplet border, and both of them were placed on the x-axis (see Figure 7b). If the direction of vibrations *a*_1_ and *a*_2_ were rotated CW, the amplitude *a*_1_ > *a*_2_ and the phase shift α = 90°, then two circular vortices appeared, but they were both rotated by a similar angle as the direction of vibrations *a*_1_ and *a*_2_ (see Figure 7c). If the phase shift α < 90°, then the more or less circular shape of the vortices was changed to an elliptical shape (see Figure 7d).

The next type of vortices presented the disturbance, and prevented the disc rotation around the pillar with micro-sized diameter. These disturbing vortices are presented in Figure 8. Four relatively large vortices were presented if the first mechanical amplifier was disconnected mechanically (see Figure 8a). The size of three vortices out of four can be reduced by applying the *a*_1_ >> *a*_2_ (see Figure 8b). The same four vortices can be rotated if the direction of the larger vibration is rotated. If the second amplifier was disconnected mechanically, then the position of vortices was rotated 90° CW (see Figure 8c). The strange combination of two large elliptical vortices appeared when the opposite direction of water stream symmetrical to the pillar appeared due to the vibrations of the glass-support plate (see Figure 8d). Both vortices remained unchanged even if the pillar was removed. If the angle of amplitudes of vibrations *a*_1_ and *a*_2_ were rotated, then both vortices also rotated around the pillar. If the amplitudes of vibrations *a*_1_ and *a*_2_ were not equal, then one of the elliptical vortices was smaller.

Whenever the vortices of the shapes described by Figure 7a–d were produced (different shapes of large vortices centred to the pillar, even if there existed a small parasitic vortex nearby the droplet border), then the micro-sized disc (rotor of the motor) rotated around the pillar. The most frequent situation of actuating vortices had the shape described by Figure 7d. Of course, the angle of the dominant (larger) vibration direction was able to rotate the elliptic vortex by 360° around the pillar. The situation with four vortices prevented rotation of the disc (Figure 8a–c) and were unusual. The four vortices were presented only when one of the mechanical amplifiers was disconnected mechanically from the glass-supported plate (broken glue bond). The most frequent disturbance, which stopped the micro-sized disc rotation completely, was the combination of two large elliptical vortices (Figure 8d). They appeared and disappeared during the change of vibration frequency. In fact, it was evident that, at most frequencies of vibrations (1 Hz–1000 Hz), two vortices, due to vibrations of the glass-support plate, became dominant over the vortices induced by the pillar. Only when resonant peaks of amplitude amplifications at certain frequencies of both mechanical amplifiers are high enough and angle shift 30° < α < 120°, then the disturbance with two large elliptical vortices was replaced by the large vortex around the pillar, which was able to rotate the disc. Usually, 2–4 such very narrow bands (20–40 Hz) of frequencies in the frequency range 1 Hz–1000 Hz were found where the disc rotated. If the angle shift was 180° < α < 225° then the rotation of the water stream around the pillar changed from the CW direction to a CCW direction. This fact was used to change the rotation direction of the motor rotor.

### 3.2. Rotating of Micro-Sized Disc around the Pillar in the Water Droplet

The electronic amplifier and sine/cosine signal generator allowed us to change the frequency of piezoelectric voltage input from 1 Hz to 1000 Hz, to change the amplitudes *a*_1_ and *a*_2_ of the input sine signal from 0% to 100% (−10 V to +115 V), and to change the shift phase between both signals between 0° and 180°. Both amplitudes *a*_1_ and *a*_2_, shift angle *α* and the speed of the rotor rotation were measured with the method presented in Section 2.3.5, and are presented in Figure 9. These measurements were made to find the perspective regions of frequencies where the rotor rotated around the pillar. The frequency regions of rotor rotation in the CW direction were: 80–100 Hz, 100–140 Hz, 160–240 Hz, 300–340 Hz and 500–560 Hz. Only one frequency region of rotor rotation in the CCW direction was found between 360 to 400 Hz (see Figure 9c). Two frequency regions were investigated more rigorously: 500–560 Hz and 360–400 Hz where the highest rotational velocity of the rotor in both directions was measured.

### 3.3. Controlling of Micro-Sized Disc Rotation Speed and Direction around the Pillar

The relationship between the amplitudes *a*_1_ and *a*_2_ on one side and phase shift *α* on the other side gave us the understanding of how the amplitude of rotational speed and the direction of rotational speed was formed and changed. If amplitudes of vibrations *a*_1_ and *a*_2_ were higher than approximately 9 µm (Figure 9a) at the same frequency bandwidth (540 Hz–580 Hz), and the phase shift was 45°< α < 135° (Figure 9b), then the high speed rotation above 25 rad/s was generated in the CW direction. If the phase shift was 225°< α < 275°, then the rotational speed of the rotor had a CCW direction. Thus, the change of the direction of the motor rotor was done simply by changing the frequency from 540 Hz to 380 Hz, or vice versa.

The selected frequencies of the rotated rotor from Figure 9c with the highest rotational speeds for both directions of rotations (CW-frequency region 536 Hz to 550 Hz and CCW-frequency region 380 Hz to 394 Hz) were measured more rigorously, with 20 measurements per measurement point. The average values and their standard deviations were calculated from these 20 measurements for every measurement point (Figure 10). The measurement points were measured every 2 Hz.

As mentioned before, the direction of the rotor rotation could be changed simply by changing the frequency of both sine and cosine input signals from the range of the frequency bandwidth 382 Hz to 392 Hz (CCW direction) to the frequency bandwidth 538 Hz to 548 Hz (CW direction). This was not all, as the amplitude of rotation could be changed continuously in the CCW direction of the rotor rotation from the minimal value, approximately −20 rad/s to 0 rad/s, by changing the frequency continuously from 382 Hz to 394 Hz. Similarly, the amplitude of rotation could be changed in the CW direction, from minimal value 0 rad/s to approximately 26 rad/s, by reducing the frequency of sine input signals from 550 Hz to 536 Hz, again continuously.

Another way to change the amplitude of the rotor of motor rotation velocity is described in Figure 11. This changed the velocity of the rotor from its max value of 25.88 rad/s to 0 rad/s when both amplitudes *a*_1_ and *a*_2_ were reduced simultaneously from 100% (*a*_1_ = 19 µm; *a*_2_ = 17 µm) to 0% at frequency 538 Hz. The angular speed of the rotated rotor was measured by 20 measurements per measurement point. The average values and their standard deviations were calculated from them for every measurement point.

Rotating of the microfluidic motor rotor, reducing the rotational speed and changing the direction of the rotation, is shown in Appendix A.

## 4. Discussion

### 4.1. Influence of Electronic and Mechanical Amplifier

The electrical sine signal made by the microcontroller (0–5 V) was amplified by the PA87 high voltage operational amplifier (−10 V to +115 V). Unfortunately, the amplification ratio of the electronic amplifier was frequency dependent, so the expected voltage amplitude −10 V to +115 V decreased above the frequency 100 Hz to the level of amplitudes below that necessary for water rotation around the pillar. This is the reason why the mechanical amplification was needed, with ratio 26 at frequency band 300 Hz to 600 Hz. Of course, such mechanical ratio was achieved only at two very narrow frequency bands with peaks 382 Hz and 538 Hz, but this was enough to control the direction and rotational speed of the microfluidic motor’s rotor.

The electrical and mechanical amplification, together with piezo actuators, were brought into the system of the microfluidic motor disturbances. The shapes of distorted mechanical vibrations, measured for the x-axis (blue curve) and y-axis (yellow curve) on both piezoelectric actuators not connected via spring connections to the glass-supporting plate, are shown in Figure 12. Figure 12a–d show the mechanical vibrations at frequencies 20 Hz, 100 Hz, 360 Hz and 600 Hz, respectively. The electrical signal before electrical amplification was proper sine. It was evident from Figure 12 that the expected sine shape of mechanical vibrations was not presented, but the phase shift remained at 90° for all frequencies. Therefore, the source of the phase shift (see Figure 9b) must be in the mechanical coupling of both mechanical amplifiers via spring connections to the glass-supporting plate. If the rigid connections from mechanical amplifiers to the glass-supporting plate were made rigid, then the phase shift was almost not presented, but the mechanical amplification ratio was lower than 2. Such a small mechanical amplification was not enough to rotate the water in the droplet around the pillar.

Mechanical coupling was the key problem to getting the repeatability of all results, especially the rotor rotation of the microfluidic motor. If the pillar was glued in the intersection of lines *c* and *d* (see Figure 13) and spring connections of both mechanical amplifiers were glued exactly at the middle of the rectangle (glass supported plate) edges a and b, then the mechanical coupling was the smallest. If the places of the pillar or the spring connections glued to the glass-supporting plate were changed, then the amplitude and phase shift characteristic vs. frequency changed dramatically with smaller amplitudes and phase shifts. Another problem was also the mounting of the piezoelectric actuators into the mechanical amplifiers. If the fastening screws (see Figure 1) were tightened too much, then the amplification ratio 26 was not reached, and vice versa, if the fastening screws were tightened insufficiently, then the more or less sine shape of the signal changed into a square shape signal with, again, lower amplification ratio.

### 4.2. Using Different Liquids for Rotation

Equations (2–9) showed that decreasing of the kinematic viscosity factor η increased the rotational speed of the water stream around the pillar. Two experiments were performed with two different liquids: Mercury (η= 0.114 × 10^−6^ m^2^/s), and metallic alloy Galinstan (η= 0.215 × 10^−6^ m^2^/s) with lower kinematic viscosity then water. Both metals were able to make droplets with a diameter smaller than 1 mm, but, surprisingly, we were unable to make a small droplet around the pillar as we had done with the water droplet. The smallest diameter of a mercury droplet around the pillar was huge, around 8 to 9 mm. In the case of Galinstan, the problem with the huge droplet around the pillar was the same. The diameter was about 1 mm smaller. We also tried with other materials instead of the glass-supporting plate and pillar (silicon, aluminium, gold, etc), but without success in reducing the size of the metallic liquid droplet. We performed the experiments successfully with rotating the metallic liquid, but the huge size of the droplet made the experiments not interesting for our purposes.

### 4.3. Estimation of Motor Maximum Torque and Load Inertia of the Microfluidic Motor

The maximum torque of our microfluidic motor was estimated with the next experiment: First, the time needed to change the rotational speed Δ*T_Sf_* for an unloaded motor rotor (Styrofoam disc) from maximal speed in a CCW direction (approximately −20 rad/s) to maximal speed in a CW direction (approximately +26 rad/s) Δω_Sf_ = 46 rad/s was measured. The time ΔT_Sf_ = 100 ms was estimated. A similar time estimation ΔT_SfAl_ = 200 ms was done for a loaded motor rotor. The load was a ring made from aluminium, with outer radius r_2Al_ = 200 µm and inner diameter r_1Al_ = 100 µm with the height h_Al_ = 140 µm. The Styrofoam disc height was increased from the usual 100 µm to h_Sf_ = 200 µm, to assure the buoyancy of a composite disc made by both discs (rings), while the inner and outer radii of the Styrofoam disc remained unchanged at r_1Al_ = 50 µm and r_2Sf_ = 200 µm, respectively. The composite disc was produced by pressing both discs together while they had been heated for a few seconds to approximately 70 °C to assure the stickiness of the discs. The scheme of the composite rotor is shown in Figure 14.

The derivation of the equation for calculating the maximal torque of the microfluidic motor is next:(11)$$\begin{array}{ll} T_{max}= & T_{SfAl}-T_{Sf}\\ & \;\;\;\;=J_{SfAl}.\frac{\Delta \omega_{SfAl}}{\Delta T_{SfAl}}+\;\omega_{SfAl}.\;B_{SfSiO2}+F_{SfSiO2}-\;J_{Sf}.\frac{\Delta \omega_{Sf}}{\Delta T_{Sf}}-\;\omega_{Sf}.\;B_{SfSiO2}\\ & \;\;\;\;-F_{SfSiO2} \end{array}$$
where $T_{SfAl}$ and $T_{Sf}$ were torques due to the composite and Styrofoam rotors, $F_{SfSiO2}$ was the Coulomb friction between the SiO_2_ pillar and Styrofoam disc (ring), $B_{SfSiO2}$ was the viscous friction between the SiO_2_ pillar and Styrofoam disc, $J_{SfAl}$ and $J_{Sf}$ were the inertia of the composite and Styrofoam rotors, respectively. Since the rotational speed of the composite rotor $\omega_{SfAl}$ and rotational speed of the Styrofoam rotor $\omega_{Sf}$ were almost equal, the terms with viscous friction were eliminated. Introducing inertia J for the shape of the rotor presented in Figure 14 gave the equation:(12)$$\begin{array}{ll} T_{max}= & \frac{\pi }{2}\{\left[\rho_{Sf}h_{Sf}\left(r_{2Sf}^{4}-r_{2Sf}^{4}\right)+\rho_{Al}h_{Al}\left(r_{2Al}^{4}-r_{2Al}^{4}\right)\right]\frac{\Delta \omega_{SfAl}}{\Delta T_{SfAl}}\\ & \;\;\;\;-\rho_{Sf}h_{Sf}\left(r_{2Sf}^{4}-r_{2Sf}^{4}\right)\frac{\Delta \omega_{Sf}}{\Delta T_{Sf}}\}=0.201\;\mathrm{pNm} \end{array}$$
where $\rho_{Sf}$ and $\rho_{Al}$ presented the density of Styrofoam and aluminium, respectively.

If the weight of the aluminium part of the rotor was increased, then the composite rotor did not reach the maximal speed in both directions anymore. In fact, if the height of the aluminium part of the composite rotor was increased to 200 µm, then (including the increase of the Styrofoam part of the rotor’s height to 300 µm), the maximal speed in both directions was reduced by twice, and the measured time variables ΔT_Sf_ = 300 ms and ΔT_SfAl_ = 600 ms gave us T_max_ = 0.047 pNm. The composite rotor was submerged deeply (approximately 90% of the height *h_Al_* + *h_Sf_*) into the water droplet. The reason for the drop of T_max_ was in the fact that the height of the water droplet was approximately 500 to 600 µm, and if the composite rotor was submerged approximately 450 µm, then the distance between the glass supported plate and rotor was decreased to 50–150 µm. This increased the friction between the water and the rotor, and also the friction between the water and the glass support plate, so much that the rotational speed of the water decreased and, therefore, lost the power to rotate the composite disc. If we had decreased the distance from the glass support plate to the composite rotor further, then the rotation of the rotor would have stopped completely.

The weight of the Styrofoam rotor was negligible in comparison with the aluminium part of the composite rotor. In fact, the inertia of the Styrofoam rotor $J_{\mathrm{Sf}}=2.5\times 10^{-17}\;\mathrm{kg}\cdot m^{2}$ (height of the disc was 100 µm) was much smaller than the inertia of the load (aluminium ring) $J_{\mathrm{Al}}=\;8.9\;\times \;10^{-16}\;\mathrm{kg}\cdot m^{2},$ which gave us the ratio between load and microfluidic rotor inertia 1:35.6, which was quite an impressive ratio.

The rotor rotational speed measurements (speed estimation) were done with the high speed camera attached on the top of the microscope. The results of the rotational speed measurements and, therefore, measurements of Δ*T_Sf_* and Δ*T_SfAl_*, were quite problematic, because the standard deviations of rotational speed reached more than 18% of the rotational speed average values, so, also, the calculated load inertia and maximal torque could have substantial measurement error, but it was of the size 0.2 pNm.

The Matlab code of the load inertia and *T_max_* calculations is attached as Appendix A.

### 4.4. Possible Improvements

We have done several experiments with rotor diameters greater than 400 µm. Of course, we needed to increase the diameter of the water droplet. We achieved larger maximum torque of the motor *T_max_* and larger inertia load with smaller maximum rotational speed of the rotor. The largest successfully performed rotor diameter was 1500 µm and water droplet diameter 3500 µm, with maximum rotational speed of 2–3 rad/s. Of course, the rotational speed would be increased if we would increase the vibration amplitudes *a*_1_ and *a*_2_, which were limited to approximately 20 µm in the present application. The maximum rotational speed in both CW and CCW directions for our rotor diameter 350–400 µm would be increased if both vibration amplitudes *a*_1_ and *a*_2_ would be increased.

The maximum rotational speed could be increased up to 60 rad/s if the rotor diameter would be decreased to only 200 µm. Further decreasing of rotor diameter would decrease the maximum rotational speed of the rotor. The last calculation was made by Equations (1–9) with parameters: The diameter of the pillar was 80 µm, frequency of vibration was 530 Hz and plural of vibrations were *a*_1_ = *a*_2_ = 18 µm. Further decreasing of the rotor diameter could be achieved by decreasing the dynamic viscosity of the liquid. In the case of using liquid metal mercury with dynamic viscosity η= 0.114 × 10^−6^ m^2^/s, the diameter of the rotor could be decreased below 30 µm according to the model described by Equations (1)–(9), with maximal rotational speed over 30 rad/sec, and with the vibration amplitudes *a*_1_ = *a*_2_
*=* 9 µm.

If the frequency of the piezo actuators would be increased to the level of a few decades of KHz, then the supporting devices’ size of the microfluidic motor (mechanical amplifier, piezoactuators, etc) could be reduced dramatically (the piezoactuator used in our application had dimensions 10 mm × 5 mm × 5 mm). For example, if the frequency would be f = 40 KHz, and amplitude of piezoactuator would reasonably be 192 nm for the piezoactuators with dimension 1 mm × 1 mm × 0.5 mm, then the diameter of the rotor could be decreased to less than 10 µm, with rotational speed in the range of 10 rad/s, according to the model described by Equations (1)–(9).

The next improvement would be the installation of quality bearings between the pillar and the rotor, so connection of the motor rotor with an axial mounted micro-sized gear-box would be possible.

The problem with evaporation of the water droplet should be minimized, or even prevented, by using low evaporation rate solvents (cyclohexanol, Eastman 2-ethylhexanol, etc).

### 4.5. Stability of Rotational Speed

Several experiments were performed to check the rotational stability at fixed experimental parameters (frequency and amplitudes of supply voltages for excitation of the piezoelectric actuators). It was discovered that the stability of rotational speed at frequency ranges 382–392 Hz (CCW direction) and 538–548 Hz was dependent on the droplet’s volume. If the volume of the droplet increased or decreased, the rotational speed decreased in both cases. If the diameter of the droplet was maintained inside the 0.75 mm to 1.2 mm range, then the rotational speed varied by about 10% in the case of maximal rotational speed in the CCW or CW directions. The highest rotational speed was achieved by droplet diameter’s size around 1 mm. It was not quite understood if this was the result of temperature change of the water droplet, or the change of the droplet’s volume. Probably, both of them were responsible for the change of speed, because the dynamic viscosity of the water η changes with the temperature, and also the volume was controlled by evaporation of water from the droplet (heating) or condensing of air moisture into the droplet (see Section 2.3.6). This effect was probably responsible for quite large standard deviations of rotational speed during the measurements presented in Figure 10 and Figure 11.

### 4.6. Endurance Test

The last test was an endurance test. We performed non-stop running of the microfluidic motor in both directions, one minute in a CW direction, and another minute in a CCW direction, with maximum rotational speed for 7–9 h, for three days consecutively. The first day, the system was working without problems. The second day, there was a failure of the system for conservation of droplet volume (see Section 2.3.6). The microfluidic motor stopped for an hour during the middle of the second day test until the electric pump for supply of cold air was replaced. The third day, the test lasted only 7 h. We had two failures per one hour each, because the glued connections between the glass supporting plate and spring connections of the mechanical amplifiers were broken. After that, we made an inspection of the complete device. Only wear of the Styrofoam rotor was observed. The inner diameter of the Styrofoam rotor was increased from approximately 100 µm to 105 µm. This is another reason why better bearings are needed for commercial use.

## Figures and Tables

**Figure 1 micromachines-10-00809-f001:**
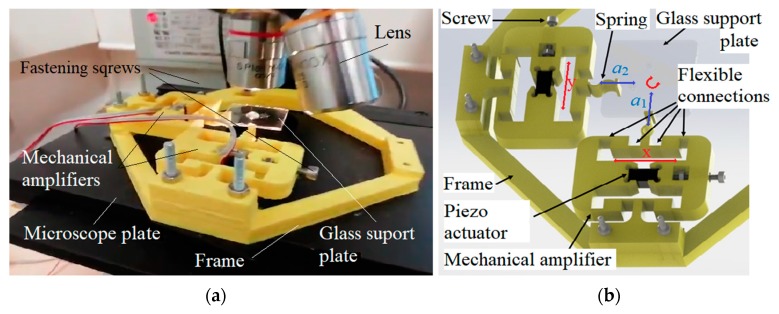
Laboratory set-up: (**a**) vibration device with the water droplet around the pillar in the middle of the support glass plate, (**b**) a detailed scheme of both mechanical amplifiers and directions of vibrations.

**Figure 2 micromachines-10-00809-f002:**
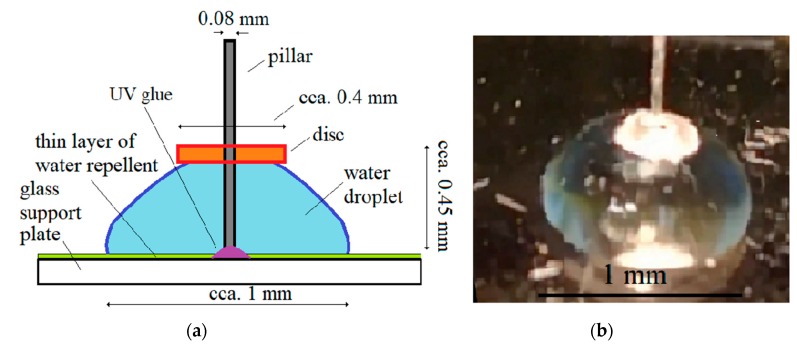
The microfluidic motor: (**a**) the scheme; (**b**) the photo.

**Figure 3 micromachines-10-00809-f003:**
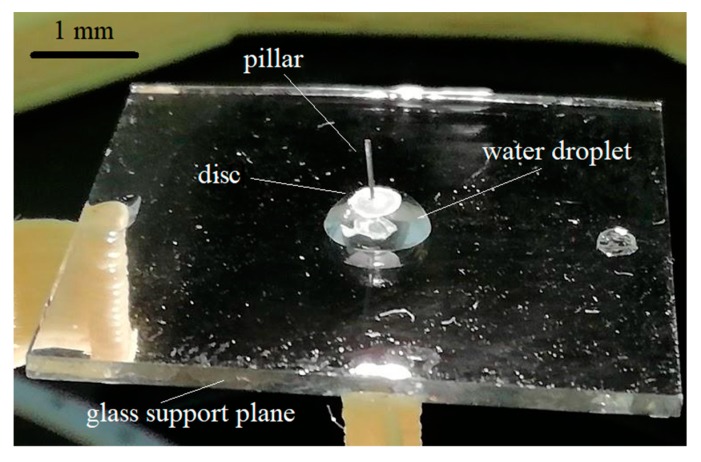
Constructed microfluidic motor on a support glass plate–disc on a drop of water with a pillar in the centre.

**Figure 4 micromachines-10-00809-f004:**
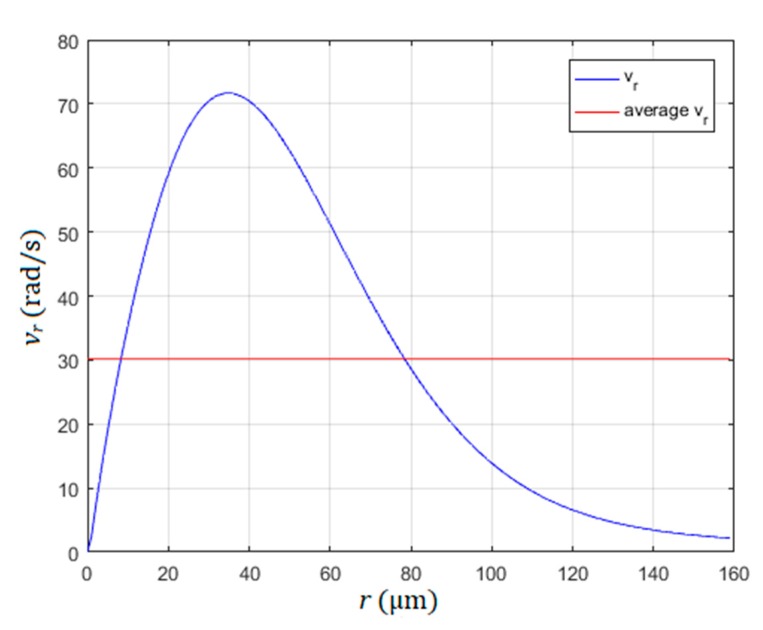
Rotational speed of water stream $v_{r}$ and its average value as a function of distance from the pillar *r*.

**Figure 5 micromachines-10-00809-f005:**
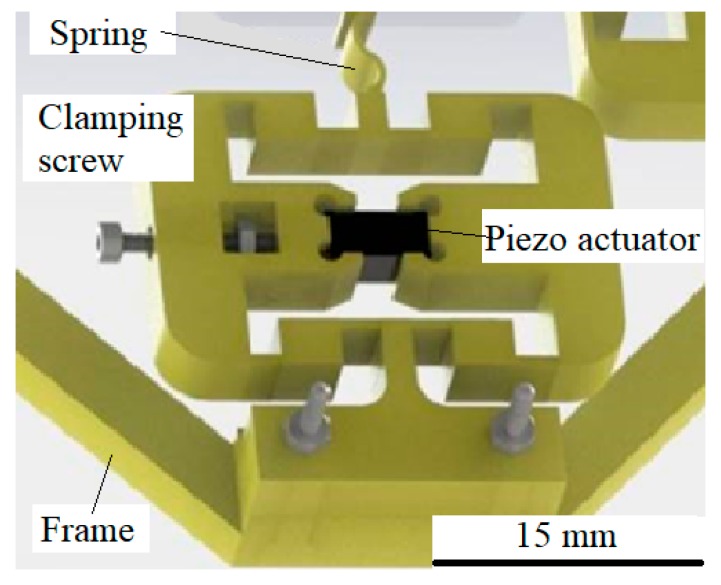
The scheme of the mechanical amplifier.

**Figure 6 micromachines-10-00809-f006:**
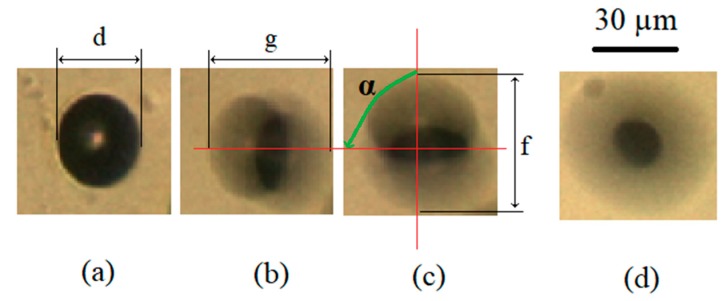
Description of the method for measurements of vibration amplitude and phase shift between directions of vibration amplitude: (**a**) both amplifiers are switched off, (**b**) only amplifier 1 is switched on, (**c**) only amplifier 2 is switched on, (**d**) both amplifiers are switched on.

**Figure 7 micromachines-10-00809-f007:**
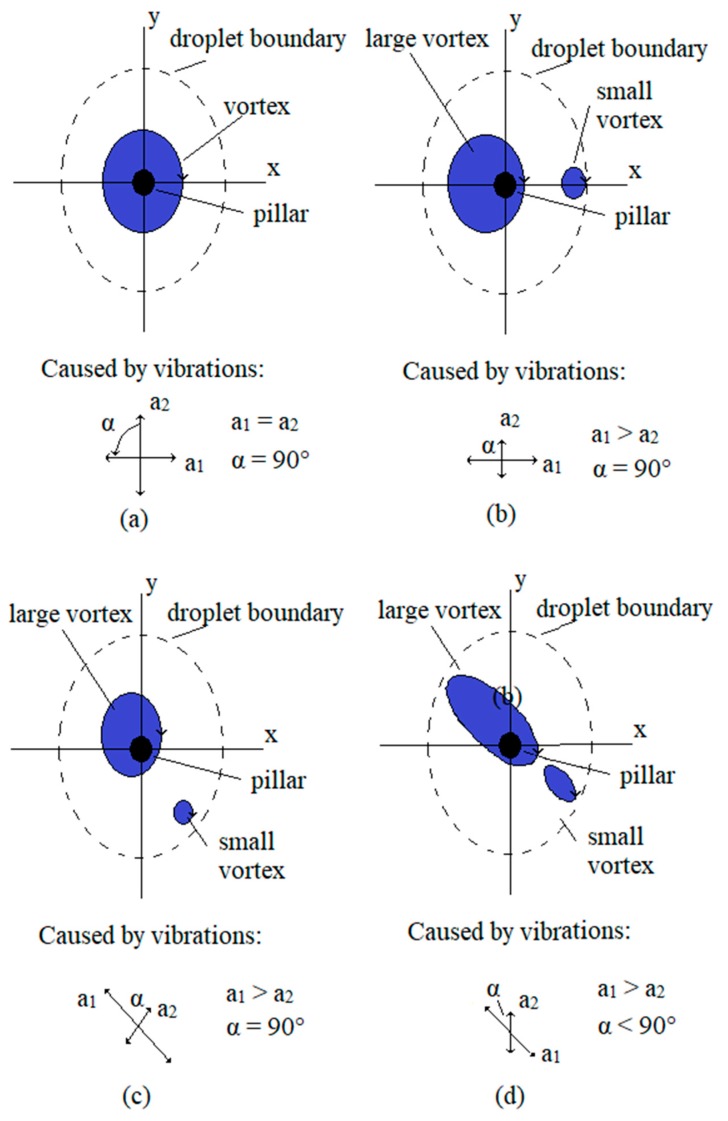
Schematic drawings of the position and shapes of actuating vortices in the water droplet due to the amplitude of vibrations *a*_1_, *a*_2_, and the phase shift between both amplitudes’ directions *α*: (**a**) single circular vortex around the pillar, (**b**) large circular vortex is centred to the pillar, both vortices are on the x-axis; (**c**) large circular vortex is centred to the pillar, both vortices are on the diagonal between both x- and y-axes; (**d**) two elliptic vortices, large elliptical vortex is centred to the pillar, both vortices are on the diagonal between x- and y-axes.

**Figure 8 micromachines-10-00809-f008:**
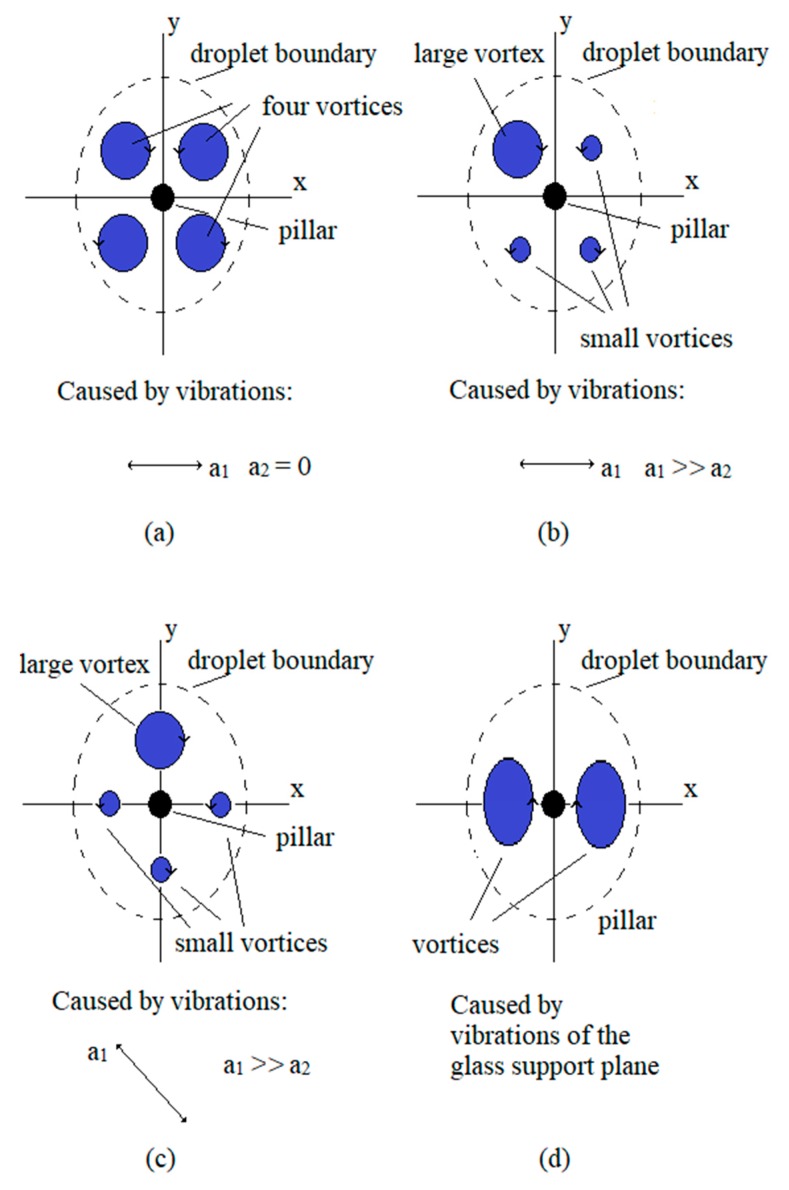
Schematic drawings of the position and shapes of disturbing vortices in the water droplet due to the amplitude of vibrations *a*_1_, *a*_2_, and the phase shift between both amplitudes’ directions *α*: (**a**) four circular vortices; (**b**) four vortices, one is large, three of them are small; (**c**) four vortices are rotated CW by 45°; (**d**) two elliptical vortices with the opposite direction of the water stream.

**Figure 9 micromachines-10-00809-f009:**
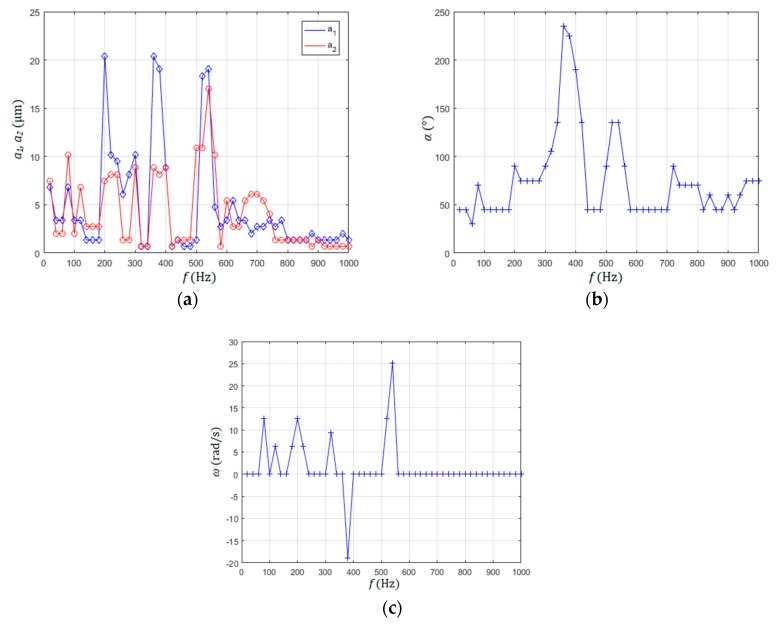
Measurements of perspective frequencies: (**a**) amplitudes *a*_1_ and *a*_2_; (**b**) phase shift *α*; (**c**) rotational speed of the rotor *ω*.

**Figure 10 micromachines-10-00809-f010:**
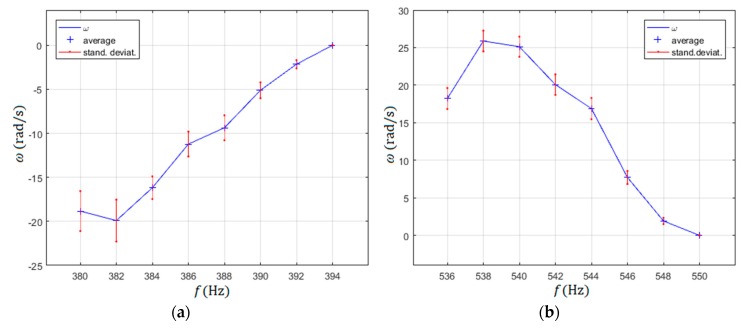
Selected frequencies of rotor rotation measured every 2 Hz: (**a**) counter clockwise (CCW) direction of rotation; (**b**) clockwise (CW) direction of rotation.

**Figure 11 micromachines-10-00809-f011:**
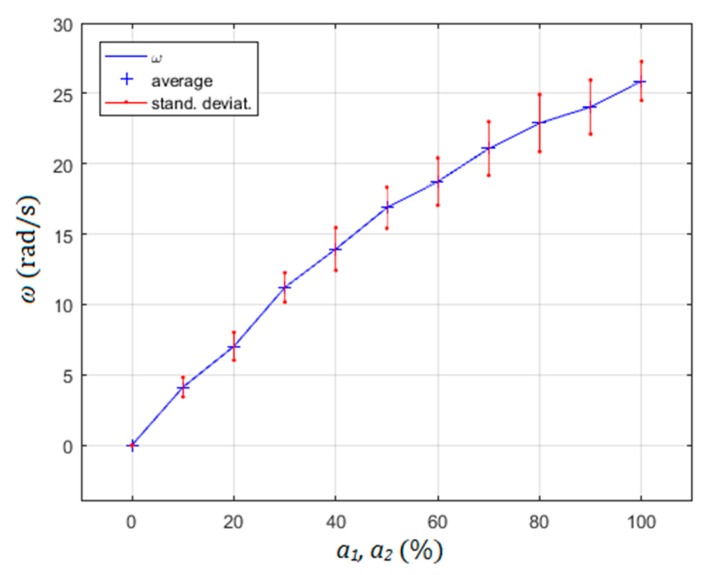
Changing the speed of rotor rotation with amplitude reduction.

**Figure 12 micromachines-10-00809-f012:**
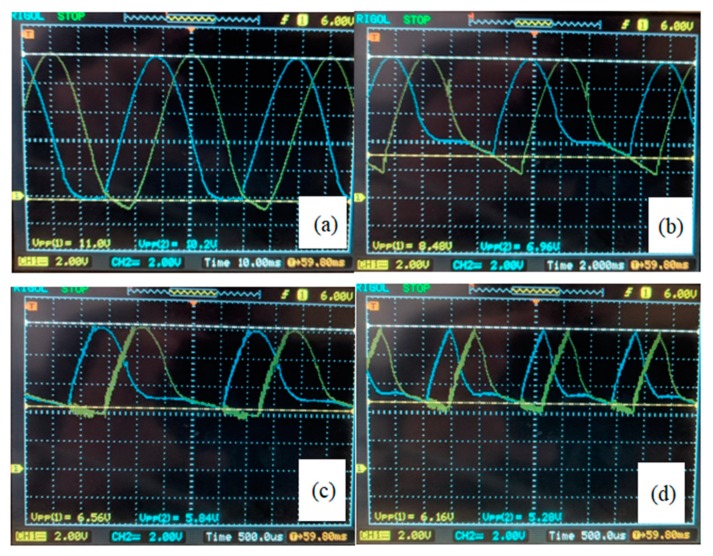
The shape of distorted vibrations of the glass support plate at different frequencies due to electrical and mechanical amplification together with piezo actuation in both x and y directions: (**a**) 20 Hz, (**b**) 100 Hz, (**c**) 360 Hz, (**d**) 600 Hz.

**Figure 13 micromachines-10-00809-f013:**
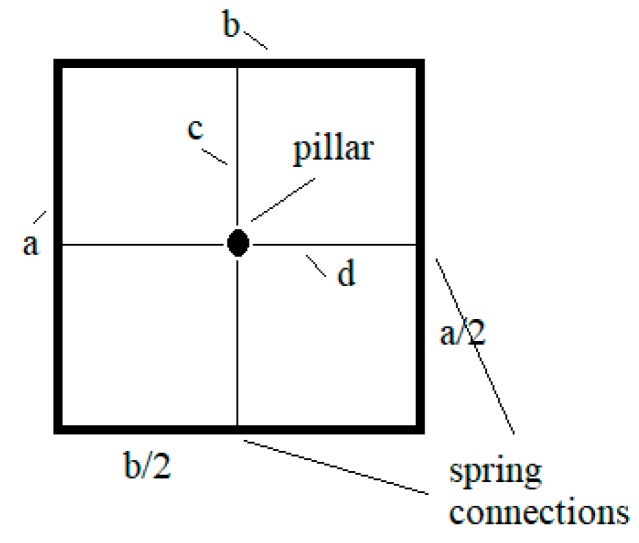
Place of the pillar and spring connections.

**Figure 14 micromachines-10-00809-f014:**
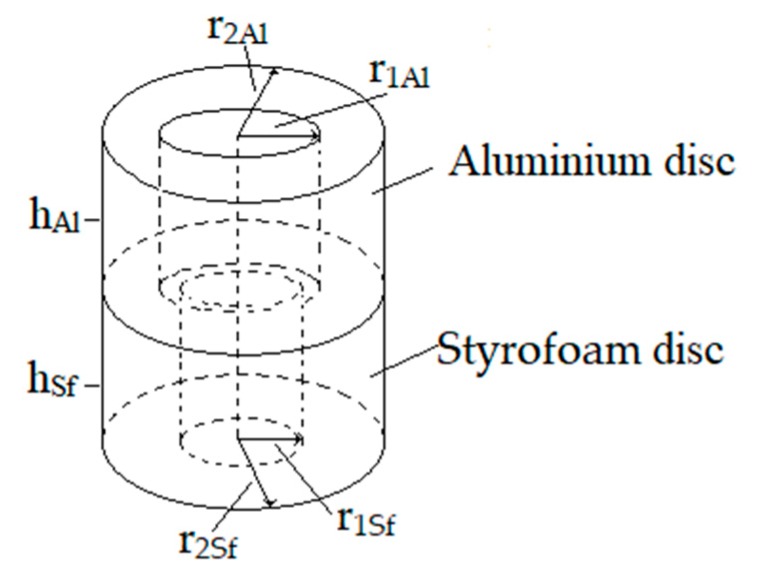
The scheme of the composite rotor.

**Table 1 micromachines-10-00809-t001:** Overview of micro motors.

Reference	Type	Rotor Size Diameter (µm)	Speed (rad/s)	Torque (pNm)	Controllabilityin Both Directions
[1]	Electrostatic, stepper and synchronous micro motor	60–120	1.2–50	order of 1	Yes
[2]	Electrostatic, salient pole side-drive and wobble micro motor	100–300	70–1500	order of 10	Yes
[3]	Electromagnetic, variable reluctance micro motor	500	up to 50	3.3 × 10^3^	Yes
[4]	Electromagnetic, variable reluctance and synchronous micro motor	1000–8000	2.1–700	3 × 10^5^–10^7^	Yes
[6]	Microfluidic driven micro motor	60–1600	390	8.7	No
[7]	Electrostatic micro motor	50	up to 21,000	–	Yes
[8,9]	Ultrasonic piezoelectric driven micro motor	700	200–500	10^7^	Yes
[12]	Laser actuated micro motor	3000	21	–	Yes
[13]	Electrowetting micro motor	2000	18	–	Yes
[14,15,16]	Surface acoustic waves micro motor	5000	235	6 × 10^4^	Yes
[17]	Microfluidic micro motor	600	125	–	No

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
