# Peer review of "A Microfluidic Rotational Motor Driven by Circular Vibrations"

_micromachines, 2019, doi:10.3390/mi10120809_

Round 1
Reviewer 1 Report
The proposed design seems to be an effective way to cope with the wear problems of the earlier rotational motor designs. The introduction is quite extensive, with enough references, well sorted and summarised. The materials and methods section provides a good overview of the proposed design, which helps understanding the results and conclusions.
However, there are several issues that must be addressed before its publication. First of all, the manuscript needs an extensive English language checking. One recursive mistake is the rotational speed unit “rd/s” which should be “rad/s”. Other typos such as: line 122 “plane” instead of “plate”, line 146 “The rotor of a microfluidic motor” instead of “The rotor of the microfluidic rotor”, line 168 “the steady term of w” instead of “the steady term of \Psi_{st}”, etc.
In section 2.3.1, the authors stateed that the water droplet flattened slightly. Adding dimensions or a scale to figure 2, or better, presenting it together with a photograph of the microfluidic motor, might be helpful.
The description of the mechanical amplifier needs more details. How are the resonant characteristics of the piezoelectric actuators (piezoelectric coefficient, resonant frequency, quality factor…)? Were the mechanical amplifiers optimised? Which were the design criteria? Was a static and vibrational mechanical analysis by finite element method performed? Authors stated that the performance of the motor was critically affected by the bolts torque and the frequency limitations of this part of the setup so, better explanation or a more in depth analysis might be helpful.
Which was the error in the determination of the vibration amplitudes produced by each amplifier (section 2.3.5)?
Due to the cooling/heating cycles to maintain the droplet volume, water might change its viscosity by more than a 30%. Was the effect of the water viscosity on the motor performance experimentally studied?
Section 3.2 (typo: should be section 3.3) needs better explanation and more comments on the presented results. Which are the authors referring to perspective frequency as? Why the mechanical amplifier 2 has lower response (figure 9a) than amplifier 1?
In section 4.2, author state that different liquids were tested with unsatisfactory results. Did they considered testing other liquids such as low evaporation solvents or low viscosity fluids?
Author Response
Comments and Suggestions for Authors
The proposed design seems to be an effective way to cope with the wear problems of the earlier rotational motor designs. The introduction is quite extensive, with enough references, well sorted and summarised. The materials and methods section provides a good overview of the proposed design, which helps understanding the results and conclusions.
Here we added the answers to the questions:
Question No.1:
However, there are several issues that must be addressed before its publication. First of all, the manuscript needs an extensive English language checking. One recursive mistake is the rotational speed unit “rd/s” which should be “rad/s”. Other typos such as: line 122 “plane” instead of “plate”, line 146 “The rotor of a microfluidic motor” instead of “The rotor of the microfluidic rotor”, line 168 “the steady term of w” instead of “the steady term of \Psi_{st}”, etc.
Answer No.1:
Thanks you very much for your accurate reading of the manuscript. We followed your remarks and we hope that there are no more typewritten errors. Also, intensive checking of grammar and other typewritten errors was done by a lector of English language (a natural speaker) throughout the text of the manuscript.
Question No.2:
In section 2.3.1, the authors stated that the water droplet flattened slightly. Adding dimensions or a scale to figure 2, or better, presenting it together with a photograph of the microfluidic motor, might be helpful.
Answer No.2:
We added the dimensions in the figure 2 a and added the photo of the droplet with the disk as Figure 2 b.
Question No.3:
The description of the mechanical amplifier needs more details. How are the resonant characteristics of the piezoelectric actuators (piezoelectric coefficient, resonant frequency, quality factor…)?
Answer No.3:
More information regarding the mechanism of mechanic amplifier, including figure 1 b, is added in section 2.2 (see lines 135 to 163). We added longitudinal piezoelectric large-signal deformation coefficient, resonant frequency, maximal push and pull forces, capacitance at 1V RMS at 1000 Hz, stiffness, etc, but we didn’t found the data regarding quality factor of used piezo stack actuator. We have sent several e-mails to the provider of piezo stack, but there have been no answer from them, yet. If we will receive the requested data from the provider, we will write the data on quality factor into the last version of the manuscript. See lines 126 to 131.
Question No.4:
Were the mechanical amplifiers optimised? Which were the design criteria? Was a static and vibrational mechanical analysis by finite element method performed?
Answer No.4:
We optimized the mechanical amplifier regarding the ability of three parameters:
Amplification ratio of amplitude of vibrations, so amplitudes a1 and a2 are both greater than 9 µm. Phase shift between 180° and 360° between the directions (vectors) of amplitudes a1 and a2, to produce CCW rotational rotor velocity. Phase shift between 0° and 180° between the directions (vectors) of amplitudes a1 and a2, to produce CW rotational rotor velocity.
We must say that it was not the easy task and we succeeded only partly, only at certain frequencies. In our case, the frequency area between 380 and 384 Hz, amplitudes a1 and a2 over 9 µm and phase shift 225°< α < 275° give us the CCW rotational velocity, while the frequency area between 536 and 550 Hz, amplitudes a1 and a2 over 9 µm and phase shift 45°< α < 135° give us the CW rotational velocity.
We must say that the optimization method was doing with trial and error. We had 3D printed more than 10 mechanical amplifiers before we were at least partly satisfied. We were changing the length and shape and also the width of the spring connections and flexible connections (see figure 1 b). The fine tuning was done by the fastening screws. We don’t exactly understand the reasons and theoretical background to make a mathematical model. We used finite element method and as well bond graph method. Both methods gave us just unusable information. We believe the reason are:
the 3D printed mechanical internal honeycomb structure of mechanical amplifiers, the quality of 3D printing was not always the same due to temperature fluctuation during the process of 3D printing, and the 3D printed internal structure partly behaved as composite material.
In fact, there were so many uncertainties regarding the mathematical modeling, so we decided not to mention this in the manuscript. It will be our goal for the future.
Question No.5:
Authors stated that the performance of the motor was critically affected by the bolts torque and the frequency limitations of this part of the setup so, better explanation or a more in depth analysis might be helpful.
Answer No.5:
As was mentioned in previous Answer No. 4 we done a lot of measurements described by figures 9, 10 and 11. We manually tighten the screws several times to find the optimal solution (calibrating) regarding amplification ratio and phase shift. The method of measurements presented by figures 9, 10 and 11 are practical measurements to understand how some parameters (spring length and shape, width of the spring connections, position of the fastening screw) affect the amplitudes a1 and a2 and also the shift angle α.
Question No.6:
Which was the error in the determination of the vibration amplitudes produced by each amplifier (section 2.3.5)?
Answer No.6:
Amplitudes a1 and a2 were calculated from measured values d, g and f (see figure 6). We added the sentence in the end of section 2.3.5: “The measurement error during the measurements of values d, g and f were about 1µm (the resolution of optical microscope) while the shift angle α had a maximal error about 5°.”
Question No.7:
Due to the cooling/heating cycles to maintain the droplet volume, water might change its viscosity by more than a 30%. Was the effect of the water viscosity on the motor performance experimentally studied?
Answer No.7:
We performed some additional measurements at fixed frequency and amplitudes of supply voltages of both piezoelectric actuators and, indeed, we discovered some dependency of rotational speed versus volume change and temperature change. So, we added a new section 4.5, where this effect was explained and discussed.
Question No.8:
Section 3.2 (typo: should be section 3.3) needs better explanation and more comments on the presented results. Which are the authors referring to perspective frequency as?
Answer No.8:
We changed the numbering of the section 3.2 into 3.3. We also added the text in the end of section 3.2 and in the paragraph (lines 399-403) in section 3.2 to clarify which perspective frequency regions and why were selected.
Question No.9:
Why the mechanical amplifier 2 has lower response (figure 9a) than amplifier 1?
Answer No.9:
Both mechanical amplifiers have not the same mechanical parameters (see frequency responses in Figures 9 a and 9 b) in spite they were printed with the same 3D printer with exactly the same 3D printing parameters at the same time and with the same program (*.stl file). As we mentioned in Answer No. 3, we believe that the quality of 3D printing was not exactly the same due to temperature fluctuation of the heated printing plate and heating of the printing nozzle during the process of 3D printing.
Question No.10:
In section 4.2, author state that different liquids were tested with unsatisfactory results. Did they considered testing other liquids such as low evaporation solvents or low viscosity fluids?
Answer No.10:
No, we didn’t, but this is a good idea for further research. We added the sentence in the end of section 4.4 (lines 594-595), where it was explained that low evaporation rate solvents should minimize the evaporation of water droplet.
Reviewer 2 Report
It's an interesting work and the results are presented properly. However, there are a few comments I'd like the authors to address before further consideration.
Compared to the other work that drives the rotors through contactless approaches, the authors rotated the disc by rotating a glass pillar that is glued on a plate and further coupling the motion to the disc through water. The design is not as flexible as the others and requires motion of bulky parts. What's the advantages of this method? Why does the phase shift change along with the change of the frequency? If the two amplifiers worked at different frequencies, the phase shift between the two amplifiers, as well as the rotation direction, keeps changing. Is that correct? What's the stability of the rotation speed for the same experimental parameters? Since the transducers has their resonant frequencies. The authors should calibrate the amplitude at different frequencies and decouple the effect of frequency and amplitude on rotation speed. Are fig. 7 and 8 simulated results or schematic drawing. The temperature of the droplet changes a lot for the purpose of maintaining the volume. How does this (change of viscosity) affect the speed of the disc. The authors described a few vortexes patterns in the droplet. Are they on the water/air interface or inside of the droplet? What's streaming profile in Z direction? In Section 4.3, the contact surface between the disc and water is different between the case with and without Al disc. It is not clear that how the authors considered the differences in their estimation.Author Response
Comments and Suggestions for Authors
It's an interesting work and the results are presented properly. However, there are a few comments I'd like the authors to address before further consideration.
Compared to the other work that drives the rotors through contactless approaches, the authors rotated the disc by rotating a glass pillar that is glued on a plate and further coupling the motion to the disc through water.
Here we added the answers to the questions:
Question No. 1:
The design is not as flexible as the others and requires motion of bulky parts.
Answer No. 1:
The microfluidic motor itself (pillar and rotor) is small and very simple to produce, while the supporting devices (mechanical amplifier with piezo actuator) used for transferring the energy via vibrations in the motor are complicated and bulky. The reason is that whenever the low frequency (lower than 1 KHz) vibrations are used to transfer the energy the dimensions of piezo actuators are in the range of 10 mm to achieve 15 µm amplitudes. The discussion, how to decrease the size of the supporting devices is added in the end of section 4.4 (lines 584-590).
Question No. 2:
What's the advantages of this method?
Answer No. 2:
We added the text in the end of section 1 which summarized the advantages of our microfluidic motor against other types of micro motors regarding the possibility to be connected to the gear box, controllability in both rotational directions, durability (wear in the bearing) and possible usage (lines 103-112).
Question No. 3:
Why does the phase shift change along with the change of the frequency? If the two amplifiers worked at different frequencies, the phase shift between the two amplifiers, as well as the rotation direction, keeps changing. Is that correct?
Answer No. 3:
Yes it is correct! The phase shift between direction of the amplitudes a1 and a2 (and also the magnitude of amplitudes a1 and a2) of mechanical vibrations changed during the change of the frequency due to mechanical properties of piezo actuators, mechanical amplifiers and mechanical coupling via glass support plate. The phase shift was measured and presented in Figure 9 and 10. The consequence of the changed phase shift is also the change of the water rotation (from CCW to CW or vise versa) in the droplet. When the phase shift was over 180° then we had CCW rotation, otherwise CW rotation. It is described in the section 3.3. Of course, the excitation voltage frequency was all the time equal for both piezo actuators, also during the change of the frequency. We added the sentence: “Both piezo actuators were excited with sine or cosine voltage signals of the same frequency, and the amplitude and the phase shift between both voltage signals could be changed independently.” in section 2.2 (lines 152-154) to clarify this fact.
Question No. 4:
What's the stability of the rotation speed for the same experimental parameters? Since the transducers has their resonant frequencies.
Answer No. 4:
We performed some measurements at fixed frequency and amplitude of supply voltages of both piezoelectric actuators and, indeed, we discovered some dependency of rotational speed versus volume change and temperature change. So, we added a new section 4.5, where this effect was explained and discussed (lines 597-612).
We also think that the transducer (piezoelectric actuator) is not the only source of the resonant peaks of rotational velocity versus frequency, but the whole chain of electronic devices (voltage amplifiers, piezoelectric actuator, mechanical amplifier and coupling of both mechanical amplifiers via the spring connections and the glass support plates). This is probably the reason why we have several peaks in Figure 9 c.
Question No. 5:
The authors should calibrate the amplitude at different frequencies and decouple the effect of frequency and amplitude on rotation speed. Are fig. 7 and 8 simulated results or schematic drawing.
Answer No. 5:
We are sorry, but we are not sure what the word “calibrate” stand for! If we understood your question correctly, we had already explained the changing of rotational speed with the change of the supply voltage amplitude at fixed (calibrated?) frequency in section 3.2 with figure 11. We also explained the change of rotational speed with the change of frequency at the fixed (calibrated?) voltage amplitudes in the same section 3.2 with figures 10 a and b.
Fig. 7 and 8 are schematic drawings based on our observations of the experiments. The photos we had made for all presented schemes were such a low quality that we decided to make schematic drawings instead. We added the information in the captions of both figures that they are schematic drawings.
Question No. 6:
The temperature of the droplet changes a lot for the purpose of maintaining the volume. How does this (change of viscosity) affect the speed of the disc.
Answer No. 6:
See the first paragraph of Answer No. 4!
Question No. 7:
The authors described a few vortexes patterns in the droplet. Are they on the water/air interface or inside of the droplet?
Answer No. 7:
We are sure that vortices are on the surface of the water droplet, because we observe the flow of small particles (dust, produced by brushing the sheet paper, lighter than water) which was floating only on the surface of the water. We also believe that similar vortices are directly beneath the droplet’s surface, for greater distance from the surface the vortices my change but they still exist. You can see the prove for this in our Supplementary material: Video S2 - rotating around pillar. The polystyrene sphere with diameter 30 µm was standing on the glass supporting plate on the bottom of the droplet at the beginning. When circular vibrations started the polystyrene sphere started to rotate around the pillar. Than it is lifted a bit in the z direction and started to rotate slower. This can be the prove that the rotation was presented also in the depth, but it changed with distance from the pillar. If it was closer to the pillar it was slower (see also fig. 4).
Question No. 8:
What's streaming profile in Z direction? In Section 4.3, the contact surface between the disc and water is different between the case with and without Al disc. It is not clear that how the authors considered the differences in their estimation.
Answer No. 8:
We didn’t measure the streaming of the water in z direction, because we believed that it was not relevant for our rotational application. But according to the information from the paper [19], from Figure 1 at the same paper [19] we can conclude that if the vibration has a vector component also into z direction, not only in x and y directions, then the rotating whirl also has the movement into the z direction towards the surface.
We didn’t considered the differences between the case with and without the Al disc. We were unable to discover exactly what was the influence on the rotational speed of the disc, which was more or less submerged in the droplet. We believe that more the disk is submerged in the droplet smaller was the acceleration when we reversed the rotational speed from CCW to CW direction. Of course, it has the influence to the calculation of the max torque. This is one of the reasons, why we called this section “estimation” and not “measurement”.
Reviewer 3 Report
This manusript "A microfluidic rotational motor driven by circular vibrations" is complete, but I think that is grammarly and poor in some sections, and the authors should revise the typos in the overall manuscript (there is many erros).
However, the content and the research presented is correct and they carefully describe the experimentally and theoretically. However, I would like to suggest the addition of table in the introduction to clarify the state of the art, and how this work contributed to improved rotational motors at the microscale and which is the status compared to the other reported works.
Also, it would be useful if author could add and schematic accompanying Figure 1 to depict how the cirucal vibrations are created and controlled (mechanical vibrationa, and location of the piezoelectric actuator), as well as improving the schematic quality of Figure 2.
Finally, I just would like to comment of how the current model/setup could be scale up (or down), as it seems that it's just valid for a very specific size and materials range.
Author Response
Here we added the answers to the questions:
Question No.1:
This manusript "A microfluidic rotational motor driven by circular vibrations" is complete, but I think that is grammarly and poor in some sections, and the authors should revise the typos in the overall manuscript (there is many erros).
Answer No. 1:
Intensive checking of grammar and typewritten errors was done by a lector of English language (a natural speaker) throughout the text of the manuscript. It was really a lot of errors and we hope all of them are corrected in the newest version of the manuscript
Question No.2:
However, the content and the research presented is correct and they carefully describe the experimentally and theoretically. However, I would like to suggest the addition of table in the introduction to clarify the state of the art, and how this work contributed to improved rotational motors at the microscale and which is the status compared to the other reported works.
Answer No. 2:
We added a Table 1, as suggested, and also how this work contributed and the status compared to the other reported work (see the end of section 1, lines 103-116).
Question No.3:
Also, it would be useful if author could add and schematic accompanying Figure 1 to depict how the circular vibrations are created and controlled (mechanical vibrations, and location of the piezoelectric actuator), as well as improving the schematic quality of Figure 2.
Answer No. 3:
Both, Figure 1 and 2 was improved. The Figure 1 was accompanied by the scheme (Figure 1 b) where the procedure of producing circular vibrations is described. Also, the text beneath the improved Figure 1 was added to explain the Figure 1 b (see lines 150-163).
Question No. 4:
Finally, I just would like to comment of how the current model/setup could be scale up (or down), as it seems that it's just valid for a very specific size and materials range.
Answer No. 4:
We added the data in the end of section 4.4 (lines 584-590) how the current lab model would be down-sized to the level of diameter of the rotor size d=30 µm according to the theoretical model described with equations (1-9) using the liquid Mercury instead of water. The up-sized dimensions of the rotor were already described in the previous version of the manuscript at the section 4.4 (lines 575-583) and are limited to the size of diameter of the rotor = 1500 µm, which was confirmed with the lab tests and predicted by the theoretical model described by equations (1-9).
Round 2
Reviewer 2 Report
The revision is satisfactory and I suggest acceptance of this work.